# Controllable Molecule Generation by Sampling in Continuous Parameter Space

## Abstract

Deep generative models have made significant strides in continuous data generation, such as producing realistic images and 3D protein conformations. However, due to the sensitivity of topological graphs to noise and the constraints of long-range discrete relationships, the generation of purely discrete data—such as topological graphs—remains a long-standing challenge, with property control proving even more elusive. In this paper, we propose a novel molecular graph generative framework, called CtrlMol, to learn the topological graphs of molecules in a differentiable parameter space. Unlike diffusion models that iteratively refine samples, CtrlMol optimizes distribution parameters at different noise levels through a pre-defined Bayesian flow. At each sampling step, we leverage a property-guided output distribution to have fine-grained control of topological graphs toward the given property. Experimental results demonstrate that CtrlMol outperforms all the competing baselines in generating natural molecule graphs. In addition, CtrlMol advances the state of the art in producing the molecules with the desired properties.

## 1 Introduction

The molecular graph generation aims to generate valid and realistic molecules with desired properties, which is a fundamental task in drug discovery. However, this problem is very challenging. On the one hand, the sample space is discrete by nature, where the data distribution is rugged and not easy to learn based on continuous assumptions. On the other hand, the molecular graphs are primarily multi-modality, including the nodes (i.e., atoms) and the edges (i.e., bonds), which are jointly dependent but follow disparate distributions. Moreover, the edges in the molecular graph are not only discrete and sparse but also obey multiple constraints in chemical valence. These distinct characteristics of the different modalities further lift the complexity of molecular graph generation.

Traditional generative approaches transform the topological molecular structures into sequences (e.g., SMILES) and formulate molecular graph generation as the sequence generation. In this way, sequential models such as LSTM and Transformer are used to produce the molecule graph in an auto-regressive manner. For example, Gómez-Bombarelli et al. (2018) learn the molecular SMILES strings with a variational auto-encoder. But the lack of explicitly syntactic constraints for molecular graphs makes them inclined to generate invalid sequences. Dai et al. (2018) and Kusner et al. (2017) impose task-specific grammars into the sequence decoder to enhance the validity of the generated sequences. As a cost of introducing the syntactic constraints, the learning of inconsecutive sequences becomes challenging.

As another alternative, graph generative networks directly output molecule graphs by iteratively adding atoms and chemical bonds. In particular, GraphAF defines a feed-forward neural network from molecular graph structures to the base distribution and generates the nodes and edges based on existing sub-graphs (Shi et al., 2020). GraphDF generates molecular graphs by sequentially sampling discrete latent variables and mapping them to new nodes and edges (Luo et al., 2021). Although graph generative models achieve impressive advancement, the graph generative process is more complicated than the sequential one due to the high degree of freedom in generative ordering. With the success of the diffusion model (DM) in producing realistic continuous images, DM is also applied to molecule generation by denoising the token embeddings in the form of molecular sequences. Runcie & Mey (2023) present a selective iterative latent variable refinement method for conditioning an existing pre-trained equivariant diffusion model toward the molecule graph generation. However,

the simplified continuity assumptions of data distribution in DM (e.g., Gaussian distribution) do not always hold in discrete data, which deepens the difficulties of multi-modality issues.

In this paper, we introduce a novel framework, CtrlMol to address the multi-modality issue in molecular graph generation. The key idea is to establish a conditional Bayes flow network with different base distributions for the sampling of the multi-modality. Concretely, CtrlMol adopts a unified probabilistic modeling formulation for different modalities in the molecular graph, including the atom identity (i.e., nodes) and the chemical bonds (i.e., edges). Second, we introduce a topological complete edge sampling strategy to address the huge complexity $O(N^2)$ in edge generation. Further, CtrlMol employs the property-guided output distribution to have fine-grained control of the topological structures. Finally, CtrlMol optimizes distribution parameters by minimizing the discrepancy between the output distribution and the true data distribution.

We evaluate CtrlMol on both unconditional and conditional molecule generation tasks. Experimental results demonstrate that CtrlMol outperforms all competing baselines in generating realistic molecular graphs. For conditional generation, CtrlMol achieves an improvement of 26% and 64% over the previous best generators in terms of QED and LogP properties, respectively. Moreover, CtrlMol could sample with any number of steps by virtue of the continuous parameter sampling space, which leads to a $100\times$ speedup with SOTA performance on conditional molecule generation.

## 2 RELATED WORK

Early methods typically adopt SMILES/SMARTS strings as a text-based representation for atom-by-atom molecule generation. By decoding the molecular strings token by token, the generative model can learn prior knowledge that encapsulates the grammar and syntax of valid SMILES (Jensen, 2019; Segler et al., 2018; Gómez-Bombarelli et al., 2018). With the development of graph neural networks (GNNs), topological graphs emerge as a more prevailing representation of molecules. For example, JTVAE (Jin et al., 2018b), GCPN (You et al., 2018), GraphAF (Shi et al., 2020), and GraphDF (Luo et al., 2021) proposed several autoregressive models to generate new nodes and edges based on the generated molecular graph.

Apart from the molecular strings (1D) and topological graphs (2D), the molecule can be also represented as the atom types and the corresponding coordinates (3D). For the generation of molecular 3D structures, diffusion models have been widely adopted. DiffSBDD (Schneuing et al., 2022) and TargetDiff (Guan et al., 2022) learn the distribution of atom types and positions from a standard Gaussian prior based via the diffusion process. DecompDiff (Guan et al., 2023) decomposes the ligand molecule and the prior into arms and scaffolds and then leverages the reverse process of diffusion to generate molecules. In fact, these diffusion-based generative methods leverage the continuous 3D coordinates to avoid the generation of the chemical bonds of atoms, which are sparse and discrete. Considering that the 3D structures of molecules are usually not with high-quality (e.g., approximated by RDKit library) and available, the generation of atoms and coordinates can hardly serve as a universal molecular graph generator, especially in the scenarios lack of 3D structures (e.g., generation toward in vivo and in vitro properties).

Recently, Bayes flow networks (BFNs) have been introduced to generate discrete structures in continuous parameter space (Qu et al., 2024; Song et al., 2023). Similar to diffusion generators, GeoBFN decomposes the molecular structure into atoms and the coordinates. Then GeoBFN adopts BFN to generate both of them and transforms the atom coordinates to build the final molecular graphs (Song et al., 2023). Different from GeoBFN, Our approach focuses on the 2D molecular graph and proposes several novel components (such as the topological complete edge sampler and property-guided output distribution) to address the multi-modality and edge sparsity issues.

## 3 METHODOLOGY

### 3.1 PROBLEM DEFINITION

We focus on the 2D molecular graphs due to the abundance of available data. The geometry of a 2D molecular graph is denoted as $\mathcal{M} = \{(\boldsymbol{V}, \boldsymbol{E})\}$, where $\boldsymbol{V} \in \mathbb{R}^{|\boldsymbol{V}| \times K_V}$ is the set of node (i.e., atoms) features and $\boldsymbol{E} \in \mathbb{R}^{|\boldsymbol{V}| \times |\boldsymbol{V}| \times K_E}$ is the set of edge (i.e., bonds) features. Each node and edge

has categorical attributes corresponding to their types. The $i$-th node is represented by a one-hot vector $\boldsymbol{v}^i \in \boldsymbol{V}$ with $K_V$ dimension, where $K_V$ represents the total number of atomic types. Similarly, the $j$-th edge $\boldsymbol{e}^j \in \boldsymbol{E}$ is represented as a one-hot vector of dimension $K_E$, with $K_E$ denoting the total number of bond types. Without loss of generality, there is an edge type to stand for no connection between two nodes. The primary goal of conditional molecular generation is to produce valid molecular structures $\mathcal{M}$ that satisfy specific properties $\boldsymbol{c}$ (e.g., logP, QED (Bickerton et al., 2012), etc.). By conditioning the generation process on these properties, we aim to explore the vast chemical space effectively, facilitating the discovery of new compounds with desired characteristics.

## 3.2 THE OVERVIEW OF BAYESIAN FLOW NETOWRKS

To effectively address the discrete nature of molecular topology, our CtrlMol model generates molecules by sampling in a continuous parameter space using the Bayesian Flow Networks (BFN) framework. The key distinction between BFN and diffusion models is that BFN refines the parameters of the data distribution rather than operating on noisy data as diffusion models do.

As illustrated in Figure 1, the generative process can be conceptualized as a message exchange between a sender and a receiver. The receiver begins with prior knowledge of the data distribution, referred to as the "input distribution" ($p_I(\mathcal{M} \mid \boldsymbol{\theta})$), where $\boldsymbol{\theta}$ is the parameters of the prior distribution for the data $\mathcal{M}$. Additionally, the receiver can estimate an "output distribution" ($p_O(\hat{\mathcal{M}} \mid \Phi(\boldsymbol{\theta}, t))$) based on the parameters $\boldsymbol{\theta}$ of the input distribution along with the processing time $t$, using a neural network to predict the data $\hat{\mathcal{M}}$.

During each message-passing step, the sender introduces noise (controlled by an accuracy parameter $\alpha$) to the origin data, creating a "sender distribution" ($p_S(\boldsymbol{y} \mid \mathcal{M}; \alpha)$), and sending a sampled data point $\boldsymbol{y}$ to the receiver. The receiver then constructs a "receiver distribution" (($p_R(\boldsymbol{y} \mid \boldsymbol{\theta}; t, \alpha)$)) by adding noise to the estimated output distribution, generating the sender sample $\boldsymbol{y}$ such that

$$p_R(\boldsymbol{y} \mid \boldsymbol{\theta}; t, \alpha) = \mathbb{E}_{\hat{\mathcal{M}} \sim p_O(\hat{\mathcal{M}} \mid \Phi(\boldsymbol{\theta}, t))} p_S(\boldsymbol{y} \mid \hat{\mathcal{M}}; \alpha). \tag{1}$$

When the sender transmits a sample from $p_S$ to the receiver, the cost incurred is the KL divergence between receiver $p_R$ and sender $p_S$. Subsequently, the receiver updates the parameters of its input distribution $p_I$ based on the received sample $\boldsymbol{y}$ by Bayesian inference, which can be expressed in closed form if the input distribution of each variable in the data is independent. Here we denote the Bayesian update function as $h$, which applies the rules of Bayesian inference to compute the updated parameters $\boldsymbol{\theta}'$, specifically, $\boldsymbol{\theta}' \leftarrow h(\boldsymbol{\theta}, \boldsymbol{y}, \alpha)$. Then the Bayesian update distribution is obtained by marginalizing out $y$ as follows:

$$p_U(\boldsymbol{\theta}' \mid \boldsymbol{\theta}, \mathcal{M}; \alpha) = \mathbb{E}_{y \sim p_S(\boldsymbol{y} \mid \mathcal{M}; \alpha)} \delta(\boldsymbol{\theta}' - h(\boldsymbol{\theta}, \boldsymbol{y}, \alpha)), \tag{2}$$

where $\delta$ is the Dirac delta distribution. The accuracy $\alpha$ has the additive property (as proven by Gra ), that is, $p_U(\boldsymbol{\theta}'' \mid \boldsymbol{\theta}, \mathcal{M}; \alpha_a + \alpha_b) = \mathbb{E}_{p_U(\boldsymbol{\theta}' \mid \boldsymbol{\theta}, \mathcal{M}; \alpha_a)} p_U(\boldsymbol{\theta}'' \mid \boldsymbol{\theta}', \mathcal{M}; \alpha_b)$. Therefore the accuracy schedule $\beta(t)$ is introduced by accumulating $\alpha$, that is

$$\beta(t) = \int_{t'=0}^{t} \alpha(t')dt'. \tag{3}$$

Then the Bayesian flow distribution can be derived by marginalizing distribution over input parameters at time $t$:

$$p_F(\boldsymbol{\theta} \mid \mathcal{M}; t) = p_U(\boldsymbol{\theta} \mid \theta_0, \mathcal{M}; \beta(t)) \tag{4}$$

Considering a sequence of $n$ steps where $n$ sender samples $\boldsymbol{y}_1, \boldsymbol{y}_2, \ldots, \boldsymbol{y}_n$ is sampled at times $t_1, t_2, \ldots, t_n$. At time step $t_i$, the sender distribution is $p_S(\cdot \mid \mathcal{M}; \alpha_i)$, where $\alpha_i = \beta(t_i) - \beta(t_{i-1})$. The receiver distribution is $p_R(\cdot \mid \boldsymbol{\theta}_{i-1}; t_{i-1}, \alpha_i)$, which is determined by the $\boldsymbol{\theta}_{i-1}$ and $t_{i-1}$ since the receiver has not received the message to update its input parameters. The input parameter sequence $\boldsymbol{\theta}_1, \boldsymbol{\theta}_2, \ldots, \boldsymbol{\theta}_n$ is updated from Bayesian update function $h$: $\boldsymbol{\theta}_i = h(\boldsymbol{\theta}_{i-1}, \mathcal{M}, \alpha(i))$. Then the $n$-step discrete-time loss $L^n(\mathcal{M})$ can be calculated by the KL divergence between the sender and receiver distributions as follows:

$$L^n(\mathcal{M}) = \mathbb{E}_{p(\boldsymbol{\theta}_1, \boldsymbol{\theta}_2, \ldots, \boldsymbol{\theta}_{n-1})} \sum_{i=1}^{n} D_{KL}(p_S(\cdot \mid \mathcal{M}; \alpha_i) \| p_R(\cdot \mid \boldsymbol{\theta}_{i-1}; t_{i-1}, \alpha_i)) \tag{5}$$

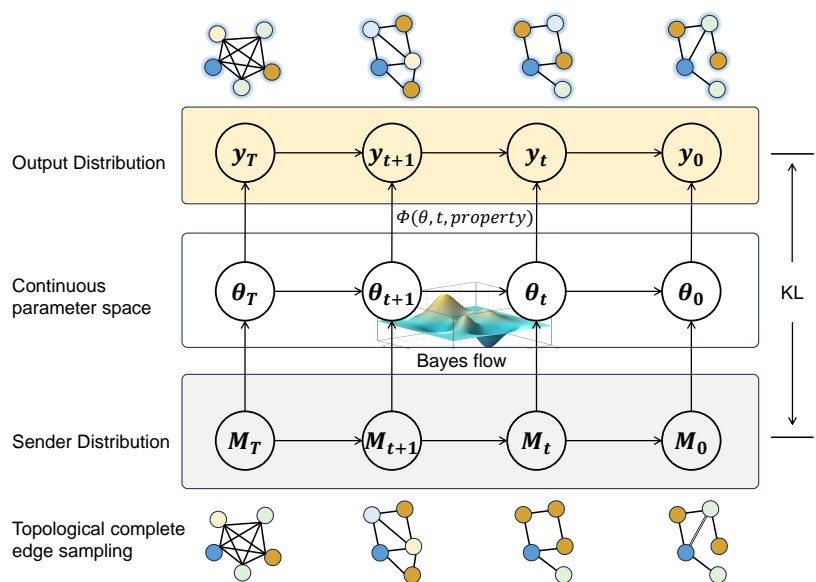

Figure 1: The generative process of the CtrlMol framework.

The summation term can be rewritten as:

$$\sum_{i=1}^{n} D_{KL} = n \times \frac{1}{n} \sum_{i=1}^{n} D_{KL} = n \times \sum_{i=1}^{n} \frac{1}{n} D_{KL} = n \underset{i \sim U\{1,n\}}{\mathbb{E}} D_{KL}. \tag{6}$$

where we abbreviate KL divergence as $D_{KL}$ and $U\{1, n\}$ is the uniform distribution over the integers from 1 to $n$. Then the discrete-time loss (Eq 5) is reformulated as

$$L^n(\mathcal{M}) = n \underset{i \sim U\{1,n\}}{\mathbb{E}} \underset{p_U(\boldsymbol{\theta}_1 | \boldsymbol{\theta}_0, \mathcal{M}; \alpha_1)}{\mathbb{E}} \underset{p_U(\boldsymbol{\theta}_2 | \boldsymbol{\theta}_1, \mathcal{M}; \alpha_2)}{\mathbb{E}} \cdots \underset{p_U(\boldsymbol{\theta}_{i-1} | \boldsymbol{\theta}_{i-2}, \mathcal{M}; \alpha_{i-1})}{\mathbb{E}} D_{KL}$$

$$= n \underset{i \sim U\{1,n\}, p_F(\boldsymbol{\theta} | \mathcal{M}; t_{i-1})}{\mathbb{E}} D_{KL}(p_S(\cdot \mid \mathcal{M}; \alpha_i) || p_R(\cdot \mid \boldsymbol{\theta}_{i-1}; t_{i-1}, \alpha_i)) \tag{7}$$

which allows us approximate $L^n(\mathcal{M})$ via Monte-Carlo sampling without summation over $n$ steps.

### 3.3 THE CTRLMOL MODEL

In this section, we will elaborate on the details of different components of CtrlMol within the field of discrete molecular graph data. Furthermore, We introduce a property-guided output distribution (as described in section 3.3.2) that allows for finer-grained control over the generation of topological structures, aiding in the production of molecules with specific properties. Since both atoms and edges in the molecules are discrete data and are represented similarly, we will primarily use nodes as examples to illustrate our model. Any differences related to edges will be explicitly noted.

#### 3.3.1 INPUT DISTRIBUTION

**Input distribution of nodes.** Given a molecular graph $\mathcal{M} = \{(\boldsymbol{V}, \boldsymbol{E})\}$, both nodes (atoms) and edges (bonds) are categorical (discrete) variables defined by their type indices. The input distribution characterized for them can be represented as a categorical distribution. It is important to note that the input distribution for each variable in a data point is independent. Take atoms $\boldsymbol{V}$ as an example, for an atom (a variable) $\boldsymbol{v}^{(i)} \in \boldsymbol{V}$, the categorical distribution with $K_V$ categories is denoted by a $K_V$-dimensional vector, $\theta^{(i)} = (\theta_1^{(i)}, \theta_2^{(i)}, \ldots, \theta_{K_V}^{(i)}) \in [0, 1]^{K_V}$, where each entry $\theta_{k_v}^{(i)}$ corresponds to the probability assigned to class $k_v$ for the variable $\boldsymbol{v}^{(i)}$. The input distribution of atoms $\boldsymbol{V}$ is then expressed as the joint probability over the true categories of all $N_v$ variables $\boldsymbol{v}^{(i)}$, i.e.,

$$p_I(\boldsymbol{V} \mid \theta) = \prod_{i=1}^{N_v} \theta_{v^{(i)}}^{(i)}. \tag{8}$$

The parameters of input distribution form a matrix $\boldsymbol{\theta} = (\theta^{(1)}, \theta^{(2)}, \ldots, \theta^{(N_v)}) \in [0, 1]^{K_V \times N_V}$. For prior $\boldsymbol{\theta}_0$, each entry is set to $\frac{1}{K_V}$ according to a uniform distribution.

**Input distribution of edges.** Similar to the node distribution, the input distribution of edges is also a categorical uniform distribution over all the edge types. Note that, the edge types and connection states can be changed during the sampling process. Therefore, we consider the possibility of no edge connection between two nodes and include a new edge type named no connections in addition to single bond, double bond, etc.

However, the parallel generation of edges $\boldsymbol{E} \in \mathbb{R}^{N_v \times N_v \times K_E}$ should be sampled from the full connections[1]. It requires the complexity of $O(N_v{}^2)$ for each of the molecular graphs, requiring huge computational resources. While the total number of edges in true data distribution is far smaller than full connections. In addition, the input distribution is the true distribution for the learning of the output distribution. With the complexity of $O(N_v{}^2)$, the output distribution should describe the probability for each entry of the full connection, leading to computational redundancy for the edge prediction.

**Topological complete edge sampling strategy.** As demonstrated in Theorem 1, the connections of a $K$-regular graph can cover any arbitrary graph with the node degree at most $K$, regardless of the node identity. In other words, if we want to generate a given topological graph with $N_v$ nodes and the degree at most $K$, the $K$-regular graph with $N_v$ nodes is complete to be the initial topological graph for sampling. Considering that the node in molecular graphs with the maximum connections is the carbon atom, we set the maximum number of the edge connections in the initial graph as $K = 4$. Formally speaking, we will randomly add several edges with "no connection" type into the given molecular graph $\mathcal{M} = \{(\boldsymbol{V}, \boldsymbol{E})\}$ to satisfy that the degree equals $K$ and then build the input distribution of edges. Note that the above topological edge sampling strategy is guaranteed by Theorem 1 and largely reduces the computational complexity to $O(K N_v)$, where $K$ is a constant.

***Theorem 1.*** *For arbitrary positive integer $N$ and $D$ ($D < N$), undirected graph $\mathcal{G} = <\mathcal{V}, \mathcal{E}>$ that has $|\mathcal{V}| = N$ and $deg(v) = D, v \in \mathcal{V}$, undirected graph $\mathcal{H} = <\mathcal{V}^H, \mathcal{E}^H>$ that has $|\mathcal{V}^H| = N$ and $deg(v) \leq D, v \in \mathcal{V}^H$, there exists at least one subgraph $\mathcal{G}' \subseteq \mathcal{G}$ such that $\mathcal{H}$ is isomorphic to $\mathcal{G}'$. (Function $deg(v)$ denotes the degree of the vertex $v$.)*

*Proof.* Our goal is to find a subgraph $\mathcal{G}' \subseteq \mathcal{G}$ that is isomorphic to $\mathcal{H}$. We employ a combinatorial argument based on the properties of regular graphs and the degrees of vertices in $\mathcal{H}$. We will take the following steps to construct a subgraph $\mathcal{G}$ of $\mathcal{G}$ using a greedy algorithm: (1) Start with an empty subgraph $\mathcal{G}'$. (2) Iterate through the vertices of $\mathcal{H}$. For each vertex $v_i$ in $\mathcal{H}$, we will select $deg(v_i)$ neighbors from $\mathcal{G}$ that are not yet included in $\mathcal{G}'$ and add them to $\mathcal{G}'$. (3) For each vertex $v_i$ in $\mathcal{H}$: Identify $deg(v_i)$ vertices from $\mathcal{G}$ that are not already included in $\mathcal{G}'$ and that can serve as neighbors. This selection is possible because $\mathcal{G}$ is $K$-regular. As long as the vertices chosen are distinct and do not exceed $K$ in degree, we can ensure that the selections are valid. (4) To ensure that the resulting subgraph $\mathcal{G}'$ is isomorphic to $\mathcal{H}$, we need to maintain the structure of $\mathcal{H}$. The selection of neighbors in $\mathcal{G}$ should respect the edges present in $\mathcal{H}$. By the property of regular graphs, since $\mathcal{G}$ has sufficient edges to accommodate the degree constraints of $\mathcal{H}$ (as $\mathcal{G}$ has $K$ edges available per vertex), we can map the edges of $\mathcal{H}$ to $\mathcal{G}'$.

After selecting neighbors for all vertices of $\mathcal{H}$, we have ensured that $\mathcal{G}'$ has $n$ vertices. The edges selected match the edges of $\mathcal{H}$ in terms of structure. Thus, we have constructed a subgraph $\mathcal{G}'$ of $\mathcal{G}$ such that $\mathcal{G}'$ is isomorphic to $\mathcal{H}$. □

### 3.3.2 PROPERTY-GUIDED OUTPUT DISTRIBUTION

The property-guided output distribution receives the parameters $\theta$ at step $t$ and the condition information $\boldsymbol{c}$ (e.g., the LogP property) to approximate the sender distribution at each step. The output distribution differs from the input distribution in that the parameters of each variable's input distribution are independent and do not incorporate contextual information (such as neighboring information). In contrast, the output distribution is formed by jointly processing the parameters of the input distributions for all variables, resulting in an interdependent distribution that integrates the contextual relationship among these variables. To achieve this, we can employ graph neural networks

---

[1]In principle, there are only $\frac{N_v \times N_v - 1}{2}$ edges needed to be determined since the edge matrix is symmetry .

(denoted by $\Phi$) to aggregate all the parameters from the input distribution and produce the output distribution.

Specifically, the input distribution parameters $\boldsymbol{\theta}$ and the process time $t$ are fed into a neural network (denoted by $\Phi$). The network then generates the output distribution parameters $\Phi(\boldsymbol{\theta}, t) = (\Phi^{(1)}(\boldsymbol{\theta}, t), \Phi^{(2)}(\boldsymbol{\theta}, t), \ldots, \Phi^{(D)}(\boldsymbol{\theta}, t))$. For a molecular graph $\mathcal{M} = \{(\boldsymbol{V}, \boldsymbol{E})\}$, the output parameters $\Phi^{(i)}(\boldsymbol{\theta}, t)$ are passed through a softmax function to produce the output categorical distribution, with the probability corresponding to the true label $v^{(i)}$ for atom $\boldsymbol{v}^{(i)} \in \boldsymbol{V}$ given by:

$$p_O^{(i)}(v^{(i)} \mid \boldsymbol{\theta}; t; \boldsymbol{c}) = (\text{softmax}(\Phi^{(i)}(\boldsymbol{\theta}, t, \boldsymbol{c})))_{v^{(i)}}. \tag{9}$$

Then the output distribution of atoms $\boldsymbol{V}$ is calculated as follows:

$$P_O(\boldsymbol{V} \mid \boldsymbol{\theta}; t; \boldsymbol{c}) = \prod_{i=1}^{N_v} p_O^{(i)}(v^{(i)} \mid \boldsymbol{\theta}; t; \boldsymbol{c}). \tag{10}$$

Note that the output distribution parameters $\Phi^{(i)}(\boldsymbol{\theta}, t)$ depend on all of $\boldsymbol{\theta}$ through the neural network $\Phi$, enabling access to contextual information.

Similarly, the output distribution of edges $\boldsymbol{E}$ can be derived in the same manner as above. Here, we utilize a graph attention neural network $\Phi$ to jointly process all the input distribution parameters. The output node embedding and edge embedding are projected to the $K_V$ and $K_E$ dimensions for the output categorical distribution of atoms $\boldsymbol{V}$ and edges $\boldsymbol{E}$, respectively.

### 3.3.3 SENDER DISTRIBUTION

The sender distribution is used to construct observation samples $\boldsymbol{y}$, which can convey information about the real data and can be considered as the messenger of the data.

Following Graves et al. (2023), the sender distribution of atom $\boldsymbol{v}^i \in \boldsymbol{V}$ could be derived with the central limit theorem, which lies in the form of

$$p_S(y_V^{(i)} \mid v_{(i)}; \alpha) = \mathcal{N}(y_V^{(i)} \mid \alpha(K_V \boldsymbol{o}_{v^{(i)}} - \mathbf{1}), \alpha K_V \mathbf{I}), \tag{11}$$

where $\mathbf{1}$ is a vector of ones, $\mathbf{I}$ is the identity matrix, and $\boldsymbol{o}_j \in \mathbb{R}^{K_V}$ is a vector defined as the projection from the class index $j$ to a length $K_V$ one-hot vector. Each entry in $\boldsymbol{o}_j$ is defined as $(\boldsymbol{o}_j)_k = \delta_{jk}$, where $\delta_{jk}$ is the Kronecker delta function. Then the sender distribution of atoms $\boldsymbol{V}$ is expressed as follows,

$$p_S(\boldsymbol{y}_V \mid \boldsymbol{V}; \alpha) = \mathcal{N}(\boldsymbol{y}_V \mid \alpha(K_V \boldsymbol{o}_V - \mathbf{1}), \alpha K_V \mathbf{I}), \tag{12}$$

where $\boldsymbol{o}_V = (\boldsymbol{o}_{v^{(1)}}, \boldsymbol{o}_{v^{(1)}}, \ldots, \boldsymbol{o}_{v^{(N_v)}}) \in \mathbb{R}^{K_V \times N_v}$.

### 3.3.4 RECEIVER DISTRIBUTION

To obtain the receiver distribution for atom $\boldsymbol{v}^i \in \boldsymbol{V}$, we substitute Eq 10 and Eq 12 into Eq 1 resulting in the following receiver distribution

$$p_R^{(i)}(y_V^{(i)} \mid \boldsymbol{\theta}; t, \alpha) = \sum_{v^{(i)}=1}^{K_V} p_O^{(i)}(v^{(i)} \mid \boldsymbol{\theta}; t) \cdot p_S(y_V^{(i)} \mid v_{(i)}; \alpha), \tag{13}$$

Then distribution distribution of atoms $\boldsymbol{V}$ is produced by

$$p_R(\boldsymbol{y}_V \mid \boldsymbol{\theta}; t, \alpha) = \prod_{i=1}^{N_v} p_R^{(i)}(y_V^{(i)} \mid \boldsymbol{\theta}; t, \alpha). \tag{14}$$

### 3.3.5 BAYESIAN UPDATES

Taking atoms $\boldsymbol{V}$ as an example, to update the input distribution parameters $\boldsymbol{\theta}_i$ given $\boldsymbol{\theta}_{i-1}$ and the atoms sample $\boldsymbol{y}_V$ drawn from $p_S(\boldsymbol{y}_V \mid \boldsymbol{V}; \alpha_i)$ (Eq 12), we utilize the Bayesian update function $h$ defined in Graves et al. (2023) as follows

$$\boldsymbol{\theta}_i \leftarrow h(\boldsymbol{\theta}_{i-1}, \boldsymbol{y}_V, \alpha_i), \tag{15}$$

$$h(\boldsymbol{\theta}_{i-1}, \boldsymbol{y}_V, \alpha_i) \stackrel{\text{def}}{=} \frac{e^{\boldsymbol{y}_V} \boldsymbol{\theta}_{i-1}}{\sum_{k=1}^{K_v} e^{\boldsymbol{y}_V^k} (\boldsymbol{\theta}_{i-1})_k}. \tag{16}$$

Once we have the Bayesian update function $h$, the Bayesian update distribution can be derived by substituting Eq 12 and Eq 16 into Eq 2, that is

$$p_U(\boldsymbol{\theta} \mid \boldsymbol{\theta}_{i-1}, V; \alpha) = \underset{\mathcal{N}(\boldsymbol{y}_V \mid \alpha(K_V \boldsymbol{e}_V - \mathbf{1}), \alpha K_V \mathbf{I})}{\mathbb{E}} \delta(\boldsymbol{\theta} - \frac{e^{\boldsymbol{y}_V} \boldsymbol{\theta}_{i-1}}{\sum_{k=1}^{K_v} e^{\boldsymbol{y}_V^k} (\boldsymbol{\theta}_{i-1})_k}) \tag{17}$$

Then we substitute Eq 17 into Eq 4, generating the Bayesian flow distribution

$$p_F(\boldsymbol{\theta} \mid \boldsymbol{V}; t) = \underset{\mathcal{N}(\boldsymbol{y}_V \mid \beta(t)(K_V \boldsymbol{e}_V - \mathbf{1}), \beta(t) K_V \mathbf{I})}{\mathbb{E}} \delta(\boldsymbol{\theta} - \frac{e^{\boldsymbol{y}_V} \boldsymbol{\theta}_0}{\sum_{k=1}^{K_v} e^{\boldsymbol{y}_V^k} (\boldsymbol{\theta}_0)_k}) \tag{18}$$

Recall that the prior is set to uniform distribution with parameters $\boldsymbol{\theta}_0 = \frac{1}{K_v}$. The above equation can be transformed into

$$p_F(\boldsymbol{\theta} \mid \boldsymbol{V}; t) = \underset{\mathcal{N}(\boldsymbol{y}_V \mid \beta(t)(K_V \boldsymbol{e}_V - \mathbf{1}), \beta(t) K_V \mathbf{I})}{\mathbb{E}} \delta(\boldsymbol{\theta} - \text{softmax}(\boldsymbol{y}_V)) \tag{19}$$

## 4 EXPERIMENT

### 4.1 DATASET

Our experiments are conducted on ZINC-250K (Irwin et al., 2012), a dataset comprising 250,000 drug-like molecules selected from the ZINC database. We performed experiments on both unconditional and conditional molecule generation tasks. For conditional molecule generation, we select two properties (drug-likeness QED and hydrophobicity LogP) to guide the generation process.

### 4.2 COMPETING MODELS

We compare our model against several state-of-art models for molecule generation. For unconditional molecule generation, our model is evaluated alongside JT-VAE (Jin et al., 2018a), GCPN (You et al., 2018), MRNN (Popova et al., 2019), GraphNVP (Madhawa et al., 2019), GRF (Honda et al., 2019), GraphAF (Shi et al., 2020), MoFlow (Zang & Wang, 2020), GraphCNF (Lippe & Gavves, 2020), and GraphDF (Luo et al., 2021). For conditional molecule generation, we compare our model with GDSS (Jo et al., 2022), DiGress (Vignac et al., 2022), GLDM, SSVAE (Mailoa et al., 2023) and ConGen Mailoa et al. (2023).

### 4.3 METRICS

We use several widely used metrics to evaluate our model on both conditional and unconditional molecular graph generation. For unconditional molecule generation, we focus on three metrics: validity, uniqueness, and novelty. Validity measures the percentage of molecules that comply with chemical valency rules. In addition, since some methods check the valency during the generation process, we also report Validity w/o check, which reflects the validity percentage when this valency correction is disabled. Uniqueness is defined as the percentage of unique molecules among the generated valid molecules. Novelty represents the percentage of the generated molecules that do not appear in the training data. For conditional molecule generation, we evaluate the performance of each model by comparing the mean absolute error (MAE) between the target properties and those of the generated molecules. All the metrics are calculated based on 10000 generated molecules.

### 4.4 IMPLEMENTATION DETAILS

Our CtrlMol uses a graph attention network to model the interdependence of the different variables and produce the output distribution. The network consists of 9 graph attention layers with 8 attention heads. Both the hidden dimension and the output dimension are 256. The node and edge features output by the graph network are then mapped to the dimensions corresponding to the number of atom types $K_V = 9$ and edge types $K_E = 5$ to generate the output distribution. The model is trained through the Adam optimizer for 1000 epochs, where the batch size is 256 and the learning rate is 0.0005.

Table 1: Random generation performance on ZINC250K dataset.

| Method | Validity | Validity w/o check | Uniqueness | Novelty |
|--------|----------|--------------------|------------|---------|
| JT-VAE | **100%** | - | **100%** | **100%** |
| GCPN | **100%** | 20% | 99.97% | **100%** |
| MRNN | **100%** | 65% | 99.89% | **100%** |
| GraphNVP | 42.6% | - | 94.8% | **100%** |
| GRF | 73.4% | - | 53.7% | **100%** |
| GraphAF | **100%** | 68% | 99.1% | **100%** |
| MoFlow | **100%** | 50.3% | 99.99% | **100%** |
| GraphCNF | 96.35% | - | 99.98% | 99.98% |
| GraphDF | **100%** | 89.03% | 99.16% | **100%** |
| Ours | **100%** | **91.6%** | **100%** | **100%** |

Table 2: Mean Absolute Error for molecular property guided generation. A lower number indicates a better controllable generation result.

| Property | QED | LogP |
|----------|-----|------|
| GDSS | 0.44 | - |
| DiGress | 0.53 | - |
| GLDM | 0.41 | - |
| HGLDM | 0.35 | - |
| SSVAE | 0.81 | 2.38 |
| ConGen | 0.82 | 2.13 |
| Ours | **0.26** | **0.77** |
| Improvement over SOTA | 25.71% | 63.85% |

## 4.5 RESULTS

**Unconditional molecule generation**    We evaluate the ability of the CtrlMol to generate realistic molecules by calculating metrics validity, uniqueness, and novelty. Table 1 shows that our model outperforms the previous methods across these metrics. While several models utilize valency checks during each sampling step to ensure the validity of generated molecules, this checking process relies on external chemical rules that the model itself does not learn, which does not accurately reflect the model's true capabilities. For instance, GCPN may achieve 100% validity through valency checking, yet it fails to capture the underlying distribution of molecules (20% validity without check).

Therefore, we focus on the validity without check, which can truly reflect the model's understanding of molecular characteristics. CtrlMol achieves the best results in validity without checking, highlighting the superiority of our model in generating realistic molecules through a one-shot sampling process without re-sampling by valency check. Additionally, CtrlMol demonstrates 100% uniqueness and novelty, highlighting its capability to produce new molecular structures that do not appear in the training data. This indicates that the model has genuinely captured the underlying distribution characteristics of the molecules, rather than merely overfitting to the training data.

**Conditional molecule generation**    For conditional molecule generation, we evaluate the CtrlMol model based on 2 properties: hydrophobicity LogP and drug-likeness QED. The model is provided with a specific property value to generate molecules that align with the desired property $c$. The property of generated molecule $\hat{c}$ is measured by RDKit. The mean absolute error (MAE) between $c$ and $\hat{c}$ is calculated to determine the relevance of the generated molecules to the conditioned property.

The results are shown in Table 2. The MSE of our model is considerably lower than that of other baseline models by an obvious margin in both the QED and LogP conditional generation tasks, suggesting that it generates molecules that more effectively align with the property conditions.

Table 3: The performance of CtrlMol with respect to different sampling steps

| Sample steps | 10 | 100 | 300 | 500 | 700 | 900 | 1000 (reference) |
|---|---|---|---|---|---|---|---|
| Validity w/o check | 85.9% | 88.9% | 89.8% | 88.0% | 87.6% | 91.5% | 91.7% |
| LogP MSE | 0.93 | 0.85 | 0.74 | 0.76 | 0.81 | 0.75 | 0.77 |

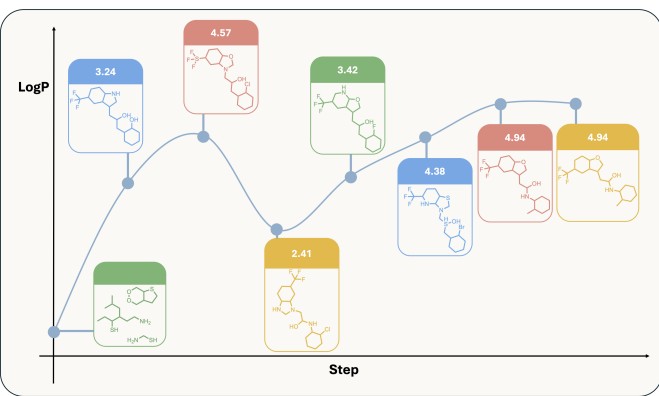

Figure 2: An example of the conditional molecule generation process.

**Few-step sampling**   Furthermore, the CtrlMol model exhibits excellent performance with a limited number of sampling steps. We show the performance in unconditional generation (validity w/o check metric) with different sampling steps in Table 3. Our model requires only 300 sampling steps to surpass the "validity w/o check" metric of GraphDF using 1000 sampling steps. Notably, with just 10 sampling steps, the model achieved a validity w/o check of 85.9%, already exceeding the results of other models with 1000 sampling steps, resulting in a $100\times$ speed-up during the sampling generation process. Similarly, in conditional generation, CtrlMol also demonstrates superiority, the property LogP of generated molecules with only 10 sampling steps is more accurate compared to other models.

**Case study**   We illustrate a conditional molecule generation process in Figure2. Given a target LogP value of 5.0, the CtrlMol model successfully generates a molecule that aligns with the desired property within 10 sampling steps. In the first step, the output samples consist of some molecular fragments. Subsequently, the model begins to output valid molecules. As the time steps progress, the logP values of the molecules gradually converge toward the target. After seven steps, the output molecules stabilize and no longer change. The logP value of the final molecule is 4.9, showcasing the model's proficiency in closely aligning with the specified property while maintaining structural validity.

## 5   CONCLUSION

Deep generative models have achieved remarkable progress in generating continuous data, such as lifelike images and 3D protein structures. Nonetheless, purely discrete data like topological molecular graphs surfer the serious issues of the multi-modality and the connection sparsity, making molecular graph generation a challenging task for the existing methods. In this paper, we introduce CtrlMol, an innovative framework for generating molecular graphs within a differentiable parameter space. In contrast to diffusion models that enhance samples through iterative refinement, CtrlMol adjusts distribution parameters at different noise levels via a non-parametric Bayesian flow. CtrlMol also integrates a topological complete edge sampling strategy and property-guided output distribution to address the above generative issue. The experimental results indicate that CtrlMol outperforms all current benchmarks in creating realistic molecular graphs, also establishing a new benchmark for generating molecules with pre-defined characteristics.

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
