# OpenReview forum: "Controllable Molecule Generation by Sampling in Continuous Parameter Space"
_ICLR.cc/2025/Conference — ICLR 2025 Conference Withdrawn Submission_

### Official Review · Reviewer_7JMd · 2024-10-28

**Soundness:** 2
**Presentation:** 2
**Contribution:** 1
**Rating:** 3
**Confidence:** 3

**Summary:**

This paper introduces CtrlMol, a molecule generative model based on Bayesian Flow Networks (BFN), which learns the topological graphs of molecules in a differentiable parameter space, and is capable of generating molecules conditionally. Experimental results demonstrate the framework’s effectiveness.

**Strengths:**

1. The paper leverages recent innovative Bayesian Flow Network architecture within the domain of topological graph generation, specifically for discrete data generation via continuous parameters. Experimental results demonstrate the model’s performance in both controlled and uncontrolled settings.

2. The paper is clearly written and well-organized.

**Weaknesses:**

1. **Limited Novelty**: A substantial portion of the paper (e.g., Section 3) is dedicated to discussing the existing BFN method or adapting BFN to topological graph data, which is already extensively covered in BFN's original paper. The main novelty, as suggested by the title and model name (i.e., “Controllable generation”), seems to be merely the inclusion of a conditioning parameter $\mathbf{c}$ within the neural network. However, this is a standard approach in other controllable generative methods (e.g., diffusion models).
2. **Insufficient Experimental evaluation**: Experiments are restricted to ZINC-250K. Evaluation on additional commonly used datasets, such as QM9 and MOSES, would provide a more comprehensive assessment of the model’s generalizability. Further, including additional metrics like FCD would strengthen the experimental rigor.
3. **Lack of Supplementary Code**: The absence of supplementary code, while not strictly required, is generally discouraged. This omission raises concerns about the reproducibility of the findings.

Overall, much of the current text would be more appropriately placed in an Appendix (which the paper currently lacks) to allow space for more comprehensive evaluations and, more importantly, for original technical or theoretical contributions beyond the adaptation of BFN to topological graph data. Based on all these observations, I find the current paper to be somewhat incomplete and in need of further substantive contributions.

**Questions:**

My primary concern relates to the paper’s level of novelty, as suggested in the Weakness section: Could you clarify the key technical problem the paper addresses beyond simply adapting BFN to topological graph data?

---

### Official Review · Reviewer_dD92 · 2024-11-01

**Soundness:** 2
**Presentation:** 2
**Contribution:** 2
**Rating:** 3
**Confidence:** 4

**Summary:**

This paper presents CtrlMol, a method based on Bayesian flow networks (BFNs) for generating the geometry of 2D molecular graphs. To tackle the computational complexity of sampling edges, which typically requires $\mathcal{O}(N^2)$ time (where $N$ is the number of nodes), the authors introduce a sampling strategy that starts with a $K$-regular graph. By setting $D$ as the maximum degree of the desired feasible molecular graph, this approach reduces the sampling complexity from $\mathcal{O}(N^2)$ to $\mathcal{O}(KN)$. Experimental results demonstrate that the proposed method achieves SOTA performance on the ZINC-250K dataset.

**Strengths:**

1. The background introduction of the paper is well-organized, clearly written, and easy to read.
2. The proposed method shows significant improvements on the ZINC-250K dataset.

**Weaknesses:**

1. **Lack of novelty**: The novelty of this work is somewhat limited, as it heavily relies on the Bayesian flow networks (BFNs) paper [1]. **The main formulation represents a straightforward application of BFNs.** Given that BFNs are inherently effective at generating discrete data, the improvements over previous works, which are presented as the primary contribution of this paper, may largely stem from the application of the BFNs framework rather than introducing significant new concepts.
2. **Paper writing**: Since BFNs is a relatively new framework with only about 22 citations, it may be challenging for readers to grasp the overall framework and implementation of BFNs. As a result, the paper lacks a detailed description of BFNs, which could hinder understanding for readers unfamiliar with the topic. It would be beneficial to include a pseudocode algorithm for both sampling and training. Also, it would also be helpful to provide a clearer, more detailed description of the objective loss function in Eq 7 to aid reader understanding (see Eq 189-190 in BFNs paper).
3. **Lack of citations**: line 119-120: BFNs, line 372: graph attention network
4. **Minors**: line 131: duplicate brackets; line 143: Gra; Eq 4: bold theta 0; footnote 1: $N \times (N-1) / 2$; Eq 11/13: subscript should be superscript; line 410: Table 1 ref*; line 465: Figure2 / Figure 2.

[1] Graves A, Srivastava R K, Atkinson T, et al. Bayesian flow networks[J]. arXiv preprint arXiv:2308.07037, 2023.

**Questions:**

Theorem 1 demonstrates that we can always obtain a desired subgraph by starting with a $K$-regular graph. However, this theorem only proves the existence of such a subgraph and does not address whether this sampling strategy complicates the sampling process. Specifically, while beginning with a $K$-regular graph can theoretically yield a feasible sample, it may be more challenging to converge to a desired sample compared to starting from a complete graph. I encourage the authors to analyze this question further, either mathematically or through empirical ablation studies. Additionally, it would be helpful if the authors could provide information on the time costs associated with applying Theorem 1 versus not applying it.

---

> ### Comment · Reviewer_dD92 · 2024-12-03
>
> As the author did not respond, I would maintain my initial rating.

---

### Official Review · Reviewer_fYrB · 2024-11-01

**Soundness:** 1
**Presentation:** 2
**Contribution:** 1
**Rating:** 3
**Confidence:** 4

**Summary:**

The paper titled "Controllable Molecule Generation by Sampling in Continuous Parameter Space" presents a molecular graph generative framework CtrlMol. It leverages Bayesian flow networks to optimize distribution parameters at different noise levels, achieving fine-grained control over the topological structures of generated molecules.

**Strengths:**

This paper is well-organized with a clear structure.

**Weaknesses:**

1. First, the paper claims that "The key distinction between BFN and diffusion models is that BFN refines the parameters of the data distribution rather than operating on noisy data as diffusion models do."(line 121.) However, despite the use of a specific optimization method, the core framework of the proposed BFN method is still based on a denoising process for learning. As mentioned earlier, the main distinction lies in the use of a particular optimization strategy, namely the design of Bayesian optimization. This does not significantly differentiates it from diffusion methods, and thus the technical contribution is quite limited.
2. The experiments in this paper are limited, as they are only tested on the ZINC dataset. Additionally, there is no mention of repeated times of the experiments, and the code is not provided, raising concerns about the reproducibility of the results.
3. There are some flaws in the writing of the paper. In many places, relevant citations and more detailed explanations are missing. For example, in the experimental section, the baselines are introduced without further explanation of how the results were obtained, and the GLDM method is not even cited. This further raises concerns about the validity of the experimental results.
4. The performance improvements in the experimental section lack theoretical support or more ablation studies. Combined with the limited innovation in the framework, I regret to say that, in its current state, the paper cannot be accepted.

**Questions:**

See Weaknesses.

---

### Official Review · Reviewer_v8U6 · 2024-11-03

**Soundness:** 2
**Presentation:** 2
**Contribution:** 3
**Rating:** 5
**Confidence:** 2

**Summary:**

The authors propose CrtlMol using Bayesian Flow Networks (BFN) to generate 2D molecular graphs, addressing the discreteness of the distribution more effectively than existing diffusion models.

**Strengths:**

1. An interesting approach using BFNs.
2. Strong empirical results on ZINC250 and good performance in conditional generation.

**Weaknesses:**

1. The empirical evaluation is somewhat limited. I would appreciate more metrics such as Frechet-ChemNet-Distance and the inclusion of other datasets (e.g., GuacaMol) as well as ablation studies on hyperparameters.
2. The paper omits comparisons with some state-of-the-art (SOTA) methods such as FreeGress [1], SyCoDiff [2], and MoLer [3].

**Questions:**

The authors argue that there is an abundance of data for 2D graphs, which justifies their focus on this setting, while noting that 3D models generally perform better (or have an easier task) due to the continuous data space. They also mention a related 3D approach, GeoBFN. In [2], the authors employ simple synthetic coordinates to enable the use of 3D models for 2D data. How would CrtlMol compare to SyCo-GeoBFN? In what scenarios would their model outperform it? I understand that a method specifically addressing discrete data distributions could offer advantages, but I would like to see a targeted discussion on which design choices enhance this work over others. Ideally, this would include a new (small) experiment.

*References*

[1] Ninniri, M., Podda, M., and Bacciu, D. Classifier-free graph diffusion for molecular property targeting. arXiv preprint arXiv:2312.17397, 2023.

[2] Ketata, Mohamed Amine, et al. "Lift Your Molecules: Molecular Graph Generation in Latent Euclidean Space." arXiv preprint arXiv:2406.10513 (2024).

[3] Maziarz, K., Jackson-Flux, H., Cameron, P., Sirockin, F., Schneider, N., Stiefl, N., Segler, M., and Brockschmidt, M. Learning to extend molecular scaffolds with structural motifs. arXiv preprint arXiv:2103.03864, 2021.

---

### Note · Authors · 2025-01-16

I have read and agree with the venue's withdrawal policy on behalf of myself and my co-authors.